# An Access Control Model for Preventing Virtual Machine Hopping Attack

**Ying Dong * and Zhou Lei**

School of Computer Engineering and Science, Shanghai University, Shanghai 200444, China; leiz@shu.edu.cn
* Correspondence: Cyvil@shu.edu.cn; Tel.: +86-1316-711-5161

**Abstract:** As a new type of service computing model, cloud computing provides various services through the Internet. Virtual machine (VM) hopping is a security issue often encountered in the virtualization layer. Once it occurs, it directly affects the reliability of the entire computing platform. Therefore, we have thoroughly studied the virtual machine hopping attack. In addition, we designed the access control model PVMH (Prevent VM hopping) to prevent VM hopping attacks based on the BLP model and the Biba model. Finally, we implemented the model on the Xen platform. The experiments demonstrate that our PVMH module succeeds in preventing VM hopping attack with acceptable loss to virtual machine performance.

**Keywords:** cloud security; virtual machine hopping; BLP model; Biba model; PVMH model

## 1. Introduction

Cloud computing is an Internet-based, emerging network computing model. It is another new computing concept after parallel computing, grid computing, and utility computing [1]. It is regarded as another revolution in the computer field. With the gradual development and advantages of cloud computing, the core technologies and applications of cloud computing have been highly valued by governments, companies, and scientific research institutions. Many IT companies such as Google [2], Amazon [3], Azure and Alibaba have taken cloud computing as an important direction for future technological innovation and invested heavily in research and development. Many countries even regard cloud computing as an important opportunity to develop and upgrade the information industry and promote the development of the information society. In the investigation report of the RightScale 2018 status, 96% of IT professionals surveyed said that their company was adopting cloud computing services, and 92% said they used public clouds. As companies move more applications to the cloud, the cloud computing market is increasingly booming. According to research firm Gartner, the public cloud market value will reach 186.4 billion US dollars in 2018, an increase of 21.4% over last year. While IT leaders decided to adopt cloud computing because of the benefits they bring, they still face a very important cloud computing challenge, one of which is security.

The virtual machine (VM) hopping attack [4,5] studied in this paper mainly involves the security between different virtual machines on the same host and the security between the virtual machine and the host. In a cloud platform, multiple virtual machines are distributed together on the same physical machine. If a virtual machine is compromised or an illegal intruder obtains the highest authority of a virtual machine by some means, there is a security risk that an illegal user uses the virtual machine as a springboard to attack other virtual machines and even attack the virtual machine manager to illegally obtain data files on the virtual machine. There are various vulnerabilities in different virtualization platforms. A Xen official announced a major security vulnerability, codenamed "Dome Breaking" (XSA-148/CVE-2015-7835). It shows that there is exploitable vulnerability in virtual machines running in the Para-Virtualized (PV) mode of the Xen platform, which is prone to virtual machine

hopping attacks and virtual machine escaping attacks. Jason Geffner of CrowdStrike found a security vulnerability related to the virtual floppy controller in the open source computer emulator QEMU, codenamed "VENOM" (CVE-2015-3456). It existed in many computer virtualization platforms (notably Xen, KVM, VirtualBox, and the native QEMU client). This vulnerability could allow an attacker to get rid of the guest identity restrictions from an infected virtual machine and likely to gain code execution rights from the host. In addition, an attacker can use it to access the host system and all virtual machines running on the host, and can elevate access permissions so that attackers can access the host's local network and neighboring systems. Another vulnerability (CVE-2018-10853) indicated that KVM 4.10 and later versions in the Linux kernel have security flaws in implementation due to the failure to detect the CPL (the privilege level of the currently executing task or program). An attacker could exploit this vulnerability to elevate privileges and cause virtual machine hopping attacks. Since the virtual machine hopping attack is a security risk of the virtualization layer, the security risks in the virtualization layer may cause the security system of the entire cloud computing platform to collapse. Therefore, if a virtual machine hopping attack occurs in the cloud platform, huge damage is brought to the entire cloud platform.

In summary, research on how to improve the security of the cloud computing platform itself and prevent malicious attacks in the cloud computing environment has theoretical and practical significance for promoting the healthy development of cloud computing platforms and their applications. The VM hopping attack is a very harmful attack method. Researchers should pay enough attention to the prevention of VM hopping attacks to make the cloud platform more stable and reliable.

In this paper, we study the related content of virtual machine hopping attacks. According to the BLP model and the Biba model, the prevent VM hopping (PVMH) model is proposed by combining the integrity and confidentiality of computer security, and strives to prevent virtual machine hopping attacks.

The rest of this paper is organized as follows: Section 2 describes several studies that are closely related to our research; Section 3 introduces some background knowledge, including virtual machine hopping attacks and access control techniques; Section 4 details the design and implementation of our proposed PVMH model; by several experiments in Section 5, we demonstrate the effectiveness of our PVMH model; and, finally, we conclude the paper and discuss future work in Section 6.

## 2. Related Works

Cloud computing and traditional IT technologies have different service models, operating modes, forms of information exchange, and technologies that provide cloud services, which makes cloud computing face different threats and risks from traditional IT technologies. Virtualization plays an important role in the construction of cloud computing. However, there are various vulnerabilities in current virtualization implementations, and the virtualization layer [6] faces various security challenges. With the help of network virtualization, a single network infrastructure can be divided into several virtual architectures. This benefits a wide range of applications, including cloud computing infrastructures. In [7], Bays et al. discussed several main challenges and threats related to the virtual network security, and presented the corresponding solutions, as well as the security aspects that had not yet been approached. In a cloud environment, security is vital to detecting intrusions into the virtual network layer. Nathiya et al. [8] proposed a hybrid intrusion detection system (H-IDS) to monitor security in a virtual network layer, but they did not deploy and verify the experiment.

The security problems in virtualization can be divided into two categories: Virtualization security risks and virtualization security attacks. Common virtualization attacks [9] include virtual machine stealing and tampering, virtual machine hopping, virtual machine escape, virtual-machine based rootkit (VMBR) attacks, and denial of service attacks. Based on the BLP model, Jiang et al. [10] proposed PVME to prevent virtual machine escape from the aspects of access control. They added two new access properties (execute (e*) and control (c*)) to the PVME model and formulated several rules for different VM states. Nguyen et al. [11] showed that the virtual switch itself can retransmit TCP

packets, which can be abused for amplification attacks by internal attackers. Rakotondravony et al. [12] presented a new classification of malware attacks in IaaS cloud environments, which helps practitioners at early stages of the design of virtual machine introspection based mitigation mechanisms by identifying relevant attacks.

Central to the cloud environment is virtualization technology, the core of which is the virtual machine (VM). Therefore, the communication capabilities between VMs are paramount. For cloud users, poor VM communication extends tenant tasks and VM lease time, eventually increasing their costs. On the other hand, the poor communication between VMs also introduces security vulnerabilities [13]. Al-Said et al. [14] outlined security challenges that exist in virtualization techniques and which are used to support several customers on one shared physical infrastructure. Ren et al. [15] analyzed the security threats and challenges virtual machines faced and presented several typical attacks to virtual machine on the Xen platform. Elmrabet et al. [16] proposed a new three-layer security architecture, which is composed of virtual switch, virtual firewall, and VLANs, to prevent attacks to virtual machines, such as sniffing, spoofing, and mac flooding. Sathya et al. [17] introduced a trusted model for VM security in cloud computing. They encrypted the VM images and used snapshot technique and a third-party monitor, all of which improves the confidentiality, integrity, and availability of VM in cloud. Mohammad-Mahdi et al. [18] presented an approach to efficiently detect side-channel attacks based on cross-VM cache, using hardware fine-grained information provided by Intel Cache Monitoring Technology (CMT) and Hardware Performance Counters (HPCs).

These studies on virtualization security have achieved relatively satisfying results in virtualization security prevention. However, when focusing on the problem of virtual machine hopping attack, these studies are relatively one-sided and do not solve this problem well. Once virtual machine hopping attacks occurs, the files on the attacked virtual machine are completely exposed to the attacker, and even the entire virtualization layer is implicated, resulting in a larger-scale leak. Therefore, preventing virtual machine hopping attacks has very high research value.

## 3. Preliminaries

### 3.1. VM Hopping

#### 3.1.1. VM Hopping Analysis

VM hopping is a common attack mode in virtualization security attacks. It means that an attacker attempts to gain access to other virtual devices on the same Hypervisor based on one virtual machine, and then attacks it. According to the implementation of virtualization, virtual machines on the same Hypervisor can communicate with each other by network connections, shared memory or other shared resources. However, it's the implementation of virtualization that results in VM hopping. VM hopping can be divided into the following two situations:

In one case, an attacker might use a malicious virtual machine to quietly access or control other virtual machines on the Hypervisor by those communication between virtual machines.

Another situation is that if an attacker from virtual machine VM1 illegally oversteps the Hypervisor layer and gains access to the host operating system, he can even destroy virtual machine VM2.

#### 3.1.2. VM Hopping Hazard

The two situations of VM hopping attacks pose a great threat to the virtualization layer and even the entire cloud platform from different aspects.

In the first situation, an attacker uses a malicious virtual machine and quietly accesses or controls other virtual machines on the same host by virtual machine communication. On the one hand, since the attacker can monitor the flow through the attacked virtual machine, he can attack the virtual machine by controlling or changing the flow. On the other hand, the attacker can modify the configuration of

the controlled virtual machine, so that the running virtual machine is forced to shut down, resulting in interruption of communication and incomplete communication. The entire attacked virtual machine is exposed to the attacker, and all files stored on it are unprotected, causing immeasurable losses to users of this virtual machine.

When the VM hopping attack succeeds, the attacker directly lands on the host by overstepping the Hypervisor layer. Inevitably, the attacker can intercept the I/O data flow of other virtual machines on this host machine, analyze and obtain relevant data of other users, and then carry out further attacks on sensitive information. If the default user, or even the administrator's basic information is modified, the host machine will be unprotected as well. What's more, if a virtual machine on the host runs as a basic service, the attacker can forcibly shut down or delete the virtual machine through Hypervisor privileges, causing the interruption of basic services and an unrecoverable disaster to the entire virtualization platform.

### 3.1.3. VM Hopping Defense

At present, VM hopping defense is mainly solved by building healthier Hypervisors and designing more robust access control policies.

(1) Build lightweight Hypervisor. In most computer systems, TCB (trusted computing base) is a combination of all the security devices that constitute a secure system. TCB, which provides security for the entire system, is highly reliable and is the basis for ensuring the safety of high-level application operations. However, the larger the TCB, the more code, the higher probability of vulnerability, and the harder it is to ensure its own credibility. Therefore, the design of lightweight Hypervisors should be as simple as possible. A lightweight Hypervisor, such as Trustvisor, Secvisor, and Cloudvisor, only retains the key feature and implements the other features in other virtualization layers. To our knowledge, ARCN, the latest lightweight Hypervisor, only has 25,000 lines of code.

(2) Design access control policy [19]. Access control is a common technical means for system security and information security, ensuring the confidentiality and integrity of data from all aspects. In the field of virtualization, the security risks are mainly caused by illegal resource access, and the design of access control is aimed at the access rights of resources between the subject and the object. Therefore, the access control policy is used to solve the related problems in the virtualized domain. Due to the more complex state transitions and hazards of virtual machine hopping attacks, a set of access control policies should be designed specifically.

In this paper, we assume the host system is trusted, and only concentrate on the access control policy of guest machines.

### 3.2. Access Control

Access control consists of three entities: The subject (sending the access request), the object (being accessed), and the security access policy (the access rules of the subject accessing the object).

Traditional access control has three access policies: (1) Discretionary Access Control [20] (DAC), which allows a subject to impose specific restrictions on access control. (2) Mandatory Access Control [21] (MAC), which does not allow subject interference to some extent. (3) Role-based access control policy [22] (RBAC), which assigns access rights according to user roles. With the rapid development of cloud computing, mobile computing, and other application scenarios, there is also an attribute-based access control [23] (ABAC).

### 3.2.1. BLP Model

The Bell-LaPadula security model [24] (BLP model), proposed by D.E. Bell and L.J. LaPadula in 1973, is a multi-level security model simulating military security strategy and is the most famous multi-level security model. The BLP model is used to control access to classified information. As the first mathematical model to formalize the security policy, it is a state machine indeed, which uses state

variables to represent the security state of the system, and state transition rules to describe the change process of the system.

The BLP model has many advantages: (1) The BLP model is one of the earliest models to describe multi-level security policy. (2) The BLP model is a strictly formalized model, and has the formalized proof. (3) The BLP model is a safe model with both discretionary access control and mandatory access control. (4) Control information can only flow from low security to high security, which meets the military department and other institution with high data confidentiality demand.

However, it also has some disadvantages: (1) Low-security information flows to high-security objects, which may damage the data integrity of high-security objects and be used by viruses or hackers. (2) As long as it is legal for the information to flow from low security to high security, it does not conform to the minimum privilege principle, no matter whether there is demand for work. (3) The BLP model focuses on confidentiality control, but lacks integrity control, so it cannot solve the problem of hidden channels, which means high-level processes can convey information to low-level processes by sharing resources.

### 3.2.2. Biba Model

The Biba model [25], proposed by K.J. Biba in 1977, is the first model that involves the integrity of computer systems. The Biba model was developed after the BLP model and was used to address application data integrity issues.

The Biba model supports five kinds of control policy: (1) Low-water-mark mandatory access control policy (LOMAC), (2) low-water-mark mandatory access control policy for object (LOMAC-O), (3) low-water-mark integrity audit policy (LO-IA), (4) ring policy (RP), and (5) strict integrity policy (SIP). Among these security policies, strict integrity is the most famous one, which is mathematically dual with the BLP security policy model. Since the strict integrity policy is most frequently used, the Biba model refers to this specific policy in most instances.

Strict integrity policy is a mathematical dual of the confidentiality strategy based on Trusted Computer System Evaluation Criteria (TCSEC). The strict integrity policy provides No Read Down (NRD) and No Write Up (NWU) characteristics. From these two characteristics, the Biba and BLP models have exactly opposite characteristics. The BLP model provides confidentiality, while the Biba model guarantees the integrity of data.

## 4. Methods

The BLP model allows information to flow from low security to high security, and prohibits the flow of information in the opposite direction. The Biba model allows information to flow from high integrity to low integrity, and prohibits the flow of information in the opposite direction. If the BLP model is directly combined with the Biba model, which implements strict integrity policy, entities at different levels are not able to communicate with each other, resulting in "information islands" in the system. Therefore, in practical application, these two models cannot be directly applied, one model has to be made with corresponding modifications to the other model.

### 4.1. PVMH Model Design

The application background of the BLP and Biba models was aimed at the traditional operating system environment. Considering that the scenario studied in this paper is a virtualization environment, the PVMH access control model is designed based on the characteristics of virtualization environments and the differences between virtualization environments and traditional operating systems. To a large extent, the PVMH model borrows from the BLP model. As a model to prevent virtual machine hopping attack, it also integrates the characteristics of the Biba model into the BLP model.

### 4.1.1. Model Elements

Basically, the PVMH model inherits most notations of the BLP model; its main model elements include: Subject, object, access attribute, access control matrix, security level, etc., as follows:

1.  Subject: The capital S represents the subject set, while the lowercase s is a single subject, namely $S = \{s_1, s_2, \ldots, s_n\}$.
2.  Object: The capital O represents the object set, while the lowercase o represents a single object, namely $O = \{o_1, o_2, \ldots, o_n\}$.
3.  Access Attribute Set $A = \{r, a, w, e\}$: The PVMH model has different attributes: Read-only ($r$), write-only ($a$), read-write ($w$), and execute ($e$).
4.  Access Matrix $M$: Each element $m_{ij}$ in $M$ represents the access permission of subject $S_i$ to object $O_j$ in current state.
5.  Security Level $R = (C, I, K)$: The capital C indicates the confidentiality level, the capital I indicates the integrity level, and the capital K indicates the security category. The PVMH model has several functions associated with security level: $f_{sc}$ for subject confidentiality, $f_{si}$ for subject integrity, $f_{sk}$ for subject security category, $f_{oc}$ for object confidentiality, $f_{oi}$ for object integrity, $f_{ok}$ for object security category, $f_{ht}$ for the highest writing-up level, and $f_{lt}$ for the lowest writing-up level. Function $f_{role}$ represents user identity, that is, whether the user is a trusted subject (Hypervisor or the privileged virtual machine) or a general subject. The notations $\geq$ and $>$ represent the partially ordered relationship of confidentiality between subject and object, while the notation $\supseteq$ represents the inclusion relationship of the security category between subject and object. The security level set $R = (R_1, R_2, \ldots, R_p)$ is a partial order set, and each item $R_i = (C_i, I_i, K_i')$ in the set represents a security level, where $C_i \in C, I_i \in I$, $K_i' \subseteq K, 1 \leq i \leq p$. $R_i$ dominates $R_j$, denoted as $R_i \geq R_j$, if and only if $C_i \geq C_j \cup I_i \geq I_j \cup K_i' \supseteq K_j'$. The functions $f_{sr}$ stands for the security levels of the subject and $f_{or}$ represents for the security levels of the object.
6.  Subject–object Security Label: The subject security label includes security level, information category (optional), and the highest writing-up level. The highest writing-up level of the subject indicates the highest security level of the object that allows the subject to perform the append or write-only access. The object security label includes security level, information category (optional), and lowest writing-up level. The lowest writing-up level of the object represents the lowest security level of the subject that allows append or write-only access to the object.
7.  Request Element $RA = \{g, r\}$: The lowercase $g$ represents a get or give request while the lowercase $r$ represents a release or rescind request.
8.  The system state set $V$ is represented by a quaternion $V = \{B \times M \times F \times H\}$, where $B = P(S \times O \times A)$ is the current access set, $b$ represents the current access set, $M$ is the access control matrix, $F$ is the access function, and $H$ is the hierarchical structure between objects, representing the subordinate relationship between objects. In object hierarchy, there is at most one node and only one parent node, and there is no ring in the structure.

### 4.1.2. Security Axioms

All security axioms of the PVMH model are named with PVME- as a prefix.

**Axiom 1 (PVMH-ds Axiom).** The PVMH-ds axiom is improved from BLP's ds-characteristic security axiom. State $v = (b \times M \times f \times H)$ satisfies the discretionary security axiom, if and only if $\forall (s, o, x) \in b, x \in M_{ij}$ is always true, where $x$ is one of four access attributes: Read-only ($r$), write-only ($a$), read-write ($w$), or execute ($e$).

**Axiom 2 (PVMH-\* Axiom).** $S'$ is a subset of $S$. A state $v = (b \times M \times f \times H)$ satisfies the *PVMH-\** axiom, if and only if for all $(s, o, x) \in b$, there exists:

$$s \in s' \Rightarrow \begin{cases} (O \in b(s:r)) & \Rightarrow & (f_{sr}(S) > f_{or}(O)) \\ (O \in b(s:a)) & \Rightarrow & (f_{sr}(S) < f_{or}(O) \text{ and } f_{ht}(S) \geq f_{or}(O) \text{ and } f_{sr}(S) \geq f_{lt}(O)) \\ (O \in b(s:w)) & \Rightarrow & (f_{sr}(S) = f_{or}(O)) \\ (O \in b(s:e)) & \Rightarrow & (S_i \in S_T) \end{cases}$$

According to BLP's ss-characteristic, the PVMH-ss-characteristic should exist in the PVMH model. However, the Hypervisor needs to communicate with the guest virtual machines, and it has full access permission to all guest virtual machines, that is, read-only (*r*), write-only (*a*), read-write (*w*), or execute (*e*). Therefore, when the Hypervisor plays the role of subject, it is in the trusted subject set $S_T$, which obviously violates the PVMH-ss-characteristic; but, for all guest virtual machines, they satisfy the PVMH-\*-characteristic, and it is easy to deduce that they also satisfy the PVMH-ss-characteristic. Therefore, due to the existence of the Hypervisor, the so-called PVMH-ss-characteristic needs to be removed from the PVME model.

4.1.3. State Transition Rules

Based on BLP security criterion and Biba model, the PVMH model improves the integrity and confidentiality of the BLP model to a certain extent. The PVMH model includes 11 state transition rules, which are expressed as $PVMH - R_i$, where $1 \leq i \leq 11$. The domain of the rule is denoted as $dom(PVMH - R_i)$. The output result is defined as the set $D = \{yes, no, ?\}$, where *"yes"* accepts the request, *"no"* rejects the request and *"?"* means the request is illegal, which does not belong to any request domain.

**Rule 1 ($PVMH - R_1$ (get-read)).** Subject virtual machine $S_i$ accesses object virtual machine $O_j$ in read-only (*r*) mode. The definition is $dom(PVMH - R_1) = \left\{ R_k \mid (g, S_i, O_j, r) \in R^{(1)} \right\}$. This pseudo code is as follow:

$$PVMH - R_1(R_k, v) = \begin{cases} (?, v) & \text{if } R_k \notin dom(PVMH - R_1) \\ (yes, (b \cup \{(S_i, O_j, r)\}, M, f, H)) & \text{if } [R_k \in dom(PVMH - R_1)] \\ & \text{and } [r \in M_{ij}] \\ & \text{and } [f_{sr}(S_i) > f_{or}(O_j) \text{ or } S_i \in S_T] \\ (no, v) & \text{otherwise} \end{cases}$$

If the decision is "yes", add a new rule that $S_i$ is allowed to access $O_j$ in read-only (*r*) mode into current access set.

**Rule 2 ($PVMH - R_2$ (get-append)).** Subject virtual machine $S_i$ accesses object virtual machine $O_j$ in write-only or append (*a*) mode. The definition is $dom(PVMH - R_2) = \left\{ R_k \mid (g, S_i, O_j, a) \in R^{(1)} \right\}$. The pseudo code is as follow:

$$PVMH - R_2(R_k, v) = \begin{cases} (?, v) & \text{if } R_k \notin dom(PVMH - R_2) \\ (yes, (b \cup \{(S_i, O_j, a)\}, M, f, H)) & \text{if } [R_k \in dom(PVMH - R_2)] \text{ and } [a \in M_{ij}] \\ & \text{and } [f_{sr}(S_i) < f_{or}(O_j)] \\ & \text{and } [f_{ht}(S_i) \geq f_{oc}(O_j)] \\ & \text{and } [f_{sc}(S_i) \geq f_{lt}(O_j)] \\ & \text{or } [S_i \in S_T] \\ (no, v) & \text{otherwise} \end{cases}$$

If the decision is "yes", add a new rule that $S_i$ is allowed to access $O_j$ in write-only or append (*a*) mode into current access set.

**Rule 3** ($PVMH - R_3$ **(get-write))**. Subject virtual machine $S_i$ accesses object virtual machine $O_j$ or $S_T$ (trusted subject) accessed object virtual machine $O_j$ in read–write ($w$) mode. The definition is $dom(PVMH - R_3) = \left\{ R_k \mid (g,\ S_i,\ O_j,\ w) \in R^{(1)} \right\}$. This pseudo code is as follow:

$$PVMH - R_3\ (R_k\ ,\ v) = \begin{cases} (?,\ v) & if\ R_k \notin dom(PVMH - R_3) \\ (yes, (b \cup \{(S_i\ ,\ O_j\ ,\ w)\}, M, f, H)) & if\ [R_k \in dom(PVMH - R_3)]\ and\ [w \in M_{ij}] \\ & and\ [f_{sr}\,(S_i) = f_{or}\,(O_j)] \\ & or\ [\ S_i \in S_T\ ] \\ (no,\ v) & otherwise \end{cases}$$

If the decision is *"yes"*, add a new rule that $S_i$ is allowed to access $O_j$ in read-write *(w)* mode into current access set.

**Rule 4** ($PVMH - R_4$ **(give-read/append/write))**. Hypervisor ($S_\lambda$) needs to set permission for subject virtual machine $S_i$ accessing object virtual machine $O_j$ in a certain mode, including read-only, write-only or read–write. The definition is $dom(PVMH - R_4) = \left\{ R_k \mid (S_\lambda,\ g,\ S_i,\ O_j,\ x) \in R^{(2)} \right\}$, $x \in A$. This pseudo code is as follow:

$$PVMH - R_4\ (R_k\ ,\ v) = \begin{cases} (?,\ v) & if\ R_k \notin dom(PVMH - R_4) \\ (yes, (b, M \backslash M_{ij} \leftarrow \{x\}, f, H)) & if\ [R_k \in dom(PVMH - R_4)] \\ & and\ [\ S_\lambda \in S_T\ ]\ and\ [\ O_j \notin O_R\ ] \\ (no,\ v) & otherwise \end{cases}$$

If the decision is "yes", add a new element that $S_i$ is allowed to access $O_j$ in x mode into access matrix.

**Rule 5** ($PVMH - R_5$ **(create-object))**. Subject $S_i$ (Hypervisor or privileged virtual machine) needs to create object virtual machine $O_j$. The definition is $dom(PVMH - R_5) = \left\{ R_k \mid (g,\ S_i,\ O_j,\ L_u) \in R^{(3)} \right\}$. The pseudo code is as follow:

$$PVMH - R_5\ (R_k\ ,\ v) = \begin{cases} (?,\ v) & if\ R_k \notin dom(PVMH - R_5) \\ (yes, (b, M, f \backslash f_o \leftarrow f_o \cup (O_j, L_u))) & if\ [R_k \in dom(PVMH - R_5)] \\ & and\ [\ S_\lambda \in S_T\ ] \\ (no,\ v) & otherwise \end{cases}$$

Notation $\leftarrow$ is assignment, which means assigning $f_o \cup (O_j,\ L_u)$ to $f_o$. Pair $(O_j,\ L_u)$ refers to mapping relation $f_o(O_j) = L_u$ while pair $(O_R,\ O_j)$ refers to another relation $H(O_R) = O_j$. Notation $f \backslash f_o \leftarrow f_o \cup (O_j,\ L_u)$ means set security level of $O_j$ to $L_u$ in security level vector (see Section 4.1.3). This expression is also used in the following rules.

If the decision is "yes", $O_j$ is created and meanwhile, the related element is added into the security level and object level.

**Rule 6** ($PVMH - R_6$ **(delete-object))**. Subject $S_i$ (Hypervisor or privileged virtual machine) needs to delete object virtual machine $O_j$ ($1 \le j \le n$), $n$ virtual machines in total). The definition is $dom(PVMH - R_6) = \left\{ R_k \mid (S_i,\ O_j) \in R^{(4)} \right\}$. The pseudo code is as follow:

$$PVMH - R_6\ (R_k\ ,\ v) = \begin{cases} (?,\ v) & if\ R_k \notin dom(PVMH - R_6) \\ \left( yes, \begin{pmatrix} (b - ACC(O_j) - OPE(S_j)), \\ M \backslash \{M_{uj} \leftarrow \varnothing, M_{ju} \leftarrow \varnothing\}, f, \\ H - (O_R, O_j) \end{pmatrix} \right) & if\ [R_k \in dom(PVMH - R_6)]\ and\ [S_\lambda \in S_T\ ] \\ & and\ [\ S_j \notin S_T\ ]\ and\ [O_j \notin O_R] \\ (no,\ v) & otherwise \end{cases}$$

In this function, $1 \le u \le n$, with $n$ virtual machines in total. Notation $ACC(O_j) = (S \times \{O_j\} \times A) \cap b$ refers to all access associated with $O_j$ in current

access set *b* while notation $OPE(S_i) = (\{S\} \times O_j \times A) \cap b$ refers to all access from $S_i$ to the deleted virtual machine in current access set *b*.

If the decision is *"yes"*, $O_j$ is deleted and, meanwhile, the related element is removed from current access *b*, access matrix *M*, and object level *H*.

**Rule 7 ($PVMH - R_7$ (rescind-read/append/write)).** The Hypervisor ($S_\lambda$) needs to revoke permission for subject virtual machine $S_i$ accessing object virtual machine $O_j$, including read-only, write-only, or read–write. The definition is $dom(PVMH - R_7) = \left\{ R_k \mid (S_\lambda, r, S_i, O_j, x) \in R^{(2)} \right\}$, $x \in A$. The pseudo code is as follow:

$$PVMH - R_7\,(R_k\,,\,v) = \begin{cases} (?,\,v) & \text{if } R_k \notin dom(PVMH - R_7) \\ \left(yes,\,(b - (S_i, O_j, x), M \backslash M_{ij} - \{x\}, f, H)\right) & \text{if } [R_k \in dom(PVMH - R_7)] \\ & \text{and } [S_\lambda \in S_T\,] \text{ and } [O_j \notin O_R] \\ (no,\,v) & \text{otherwise} \end{cases}$$

If the decision is *"yes"*, remove the element that $S_i$ can access $O_j$ in *x* mode from access matrix, and, meanwhile, remove the rule that $S_i$ can access $O_j$ in *x* mode from current access set.

**Rule 8 ($PVMH - R_8$ (modify-object-l)).** The Hypervisor needs to modify $L_u$, the security level of object virtual machine.

In the following, $PVMH - *(R_k,\,v)$ is the characteristic function, which guarantees that if it outputs "true" and state *v* satisfies the PVMH-* characteristic with respect to $S^*(S^* \subseteq S)$, the state v after this transition also satisfies the PVMH-* characteristic. The strict mathematic definition is as follows:

$PVMH - *(R_k,\,v) = \;true \Leftrightarrow$
$\forall S_\lambda \in S',\;\left[ (S_\lambda,\,O_j,\,a\,) \in b \Rightarrow L_u f_{sr}(S_\lambda)\,\&f_{sc}(S_\lambda) \geq f_{lt} \right]$
$\&\left[ (S_\lambda,\,O_j,\,w\,) \in b \Rightarrow L_u = f_{sr}(S_\lambda) \right]$
$\&\left[ (S_\lambda,\,O_j,\,r\,) \in b \Rightarrow L_u f_{sr}(S_\lambda) \right]$
$\forall O_\lambda \in O,\;\left[ (S_\lambda,\,O_j,\,a\,) \in b \Rightarrow f_{or}(O_\lambda)L_u\&f_{oc}(O_\lambda) \geq f_{ht} \right]$
$\&\left[ (S_\lambda,\,O_j,\,w\,) \in b \Rightarrow f_{or}(O_\lambda) = L_u \right]$
$\&\left[ (S_\lambda,\,O_j,\,r\,) \in b \Rightarrow f_{or}(O_\lambda)\,L_u \right]$

The definition is $dom(PVMH - R_8) = \left\{ R_k \mid (r,\,S_i,\,O_j,\,L_u) \in R^{(3)} \right\}$. The pseudo code is as follows:

$$PVMH - R_8\,(R_k,\,v) = \begin{cases} (?,\,v) & \text{if } R_k \notin dom(PVMH - R_8) \\ \left(yes,\,(b, M, f \backslash f_o \leftarrow f_o \cup (O_j, L_u), H \cup (O_R, O_j))\right) & \text{if } [R_k \in dom(PVMH - R_8)] \\ & \text{and } [S_i \in S_T\,] \text{ and } [O_j \neq O_R] \\ & \text{and } [PVMH - *(R_k,\,v) = true] \\ (no,\,v) & \text{otherwise} \end{cases}$$

If the decision is "yes", set security level of $O_j$ to $L_u$.

**Rule 9 ($PVMH - R_9$ (modify-$f_{sc}$)).** The trusted subject needs to modify the highest writing-up level for a certain subject virtual machine. The definition is $dom(PVMH - R_9) = \left\{ R_k \mid (r,\,S_i,\,O_j,\,f_{ht}) \in R^{(5)} \right\}$. The pseudo code is as follows:

$$PVMH - R_9\,(R_k,\,v) = \begin{cases} (?,\,v) & \text{if } R_k \notin dom(PVMH - R_9) \\ (yes,\,(f_{ht} = Highest)) & \text{if } [R_k \in dom(PVMH - R_9)] \\ & \text{and } [f_{role}(S_i) = admin\,] \text{and} \left[ (S_m, O_j, a) \notin b \right] \\ & \text{and} [f_{sc}(S_i) \leq Highest] \\ (no,\,v) & \text{otherwise} \end{cases}$$

Notation $f_{ht}$ refers to the highest writing-up level of the subject, while "*Highest*" refers to the highest writing-up level of the subject granted by administration. If the decision is "yes", the highest writing-up level of the subject is updated to "*Highest*".

**Rule 10** *(PVMH − $R_{10}$ (modify-$f_{oc}$)).* The trusted subject needs to modify the lowest writing-up level for a certain subject virtual machine. The definition is $dom(PVMH - R_{10}) = \left\{ R_k \mid (r,\, S_i,\, O_j,\, f_{lt}) \in R^{(5)} \right\}$. The pseudo code is as follows:

$$
PVMH - R_{10}\,(R_k\,,\, v) = \begin{cases} (?,\, v) & if\ R_k \notin dom(PVMH - R_{10}) \\ (yes, (f_{ht} = Lowest)) & if\ [R_k \in dom(PVMH - R_{10})] \\ & and\ [f_{role}(S_i) = admin\,]and\big[(S_m, O_j, a) \notin b\big] \\ & and\big[f_{oc}(O_j) \leq Lowest\big] \\ (no,\, v) & otherwise \end{cases}
$$

Notation $f_{lt}$ refers to the lowest writing-up level of the subject, while "*Lowest*" refers to the lowest writing-up level of the subject granted by administration. If the decision is "yes", the lowest writing-up level of the subject is updated to "*Lowest*".

**Rule 11** *(discretionary access control).* The access matrix allows the subject virtual machine $S_i$ to access the object virtual machine $O_j$ in $x$ mode only if $x$ is contained in both the row element of $S_i$ and column element of $O_j$ in the access control matrix. The state $v$ satisfies the discretionary security characteristic if and only if $(S_i,\, O_j,\, x) \in b \Rightarrow x \in M_{ij}$.

By setting the highest writing-up level and the lowest writing-up level that each subject can write into, the PVMH model implements stricter integrity restrictions, while keeping most security characteristics of the BLP model, so it also has higher security.

*4.2. PVMH Model Mapping*

### 4.2.1. Subject–Object Mapping

This paper takes Xen as the virtualization platform. In Xen, the privileged virtual machine is denoted as Domain0, the ordinary virtual machine is denoted as DomainU, and the virtual machine manager is denoted as Hypervisor. Domain0 and Hypervisor are the trusted subjects, which manage all virtual machines on the same host. In the BLP model, both subject and object are abstract words, while in the cloud platform system, the subject may be Hypervisor, Domain0, or DomainU, and the object may be Hypervisor, DomainU, or a specific file, memory snip, data unit, etc. When access control is Hypervisor-related, the Hypervisor is at the highest level of confidentiality and integrity.

### 4.2.2. Access Attribute Mapping

In Xen, read and write operations between guest virtual machines are accomplished through communication mechanisms. The interaction between guest virtual machines corresponds to the access properties of the PVMH model. In the PVMH model, event channels can be established and event notifications sent as long as one guest virtual machine has some access to attribute to another guest virtual machine. The corresponding operation can be found in state transitions of the PVMH model, as shown in Table 1.

**Table 1.** State transition rules.

| Virtual Machine State | Corresponding Function | Effect | Corresponding Transition Rules |
|---|---|---|---|
| virtual machine management | modifying access permission | Hypervisor modifies access permission for a certain virtual machine. | Rule 8 |
| | modifying object security level | Hypervisor modifies security level of a certain subject virtual machine. | Rule 10 |
| | modifying subject security level | Hypervisor modifies security level of a certain object virtual machine. | Rule 9 |
| | authorizing access attribute | Hypervisor grants some access attribute to a certain subject virtual machine. | Rule 4 |
| | releasing access attribute | Hypervisor revokes some access attribute from a certain subject virtual machine. | Rule 7 |
| creating a guest virtual machine | creating a subject | Hypervisor creates a subject virtual machine and sets its security level. | Rule 5 |
| | creating an object | Hypervisor creates an object virtual machine and sets its security level. | Rule 5 |
| deleting a guest virtual machine | deleting a subject | Hypervisor deletes a subject virtual machine and its related data. | Rule 6 |
| | deleting an object | Hypervisor deletes an object virtual machine and its related data. | Rule 6 |
| virtual machine communication (access to shared data, etc.) | writing an object | Subject virtual machine writes data into and reads data from object virtual machine. | Rule 2 |
| | reading an object | Subject virtual machine only reads data from object virtual machine. | Rule 1 |
| | appending an object | Subject virtual machine only writes data into object virtual machine. | Rule 3 |

### 4.2.3. Access Matrix Mapping

In our framework designed with the PVMH model, the access matrix is stored as a binary file in the virtual machine manager, while a backup file is also stored in the privileged virtual machine. Each element of the access matrix is a one-dimensional ordered tuple (SID, OID, R, A, W, Flag) where SID is the subject security ID number, OID is the object security ID number, R is read-only access attribute, A is write-only access attribute, W is read–write access attribute, and Flag is used to indicate whether the rule is valid; "1" is valid and "0" is invalid. The security ID number is set by the system administrator, and the ID number of the virtual machine in the cloud platform is allocated by the cloud system when the virtual machine starts up. For a virtual machine, both the SID and the OID are equal to its security ID number, that is, SID = OID = ID. In the cloud platform, the ID number is a 13-bit binary number, and R, A, and W are represented by 1-bit binary numbers. When R (or A or W) is set to 1, the subject has the access attribute for the object, and when the value is set to 0, the subject does not have the access attribute for the object. It can be seen that the six-element tuple has a total of 30 binary digits (13 bits for SID and OID, 1 bit for R, A, and W, and 1 bit for Flag), and the default access matrix is sorted by pair (SID, OID) in ascending order, which is very efficient for searching later.

### 4.2.4. Current Access Set

The current access set $b$ ($b \subseteq S \times O \times A$) includes all access that the subject has for the object in some certain modes. The current access set can be used to determine whether the state of the system is secure. In the PVMH model, each subject has its own current access set $b$, denoted as $b(S) \subseteq O \times A$. The elements in the object set $O$ are represented by OID, and the access attribute set $A$ includes three elements: Read-only ($r$), write-only ($a$), and read–write ($w$).

### 4.2.5. Security Level

The PVMH model has 8 confidentiality levels and 8 integrity levels. The confidentiality set is defined as $C = \{C_1, C_2, \ldots, C_8\}$, where $C_1 > C_2 > \ldots > C_8$, while the integrity set is defined as $I = \{I_1, I_2, \ldots, I_8\}$, where $I_1 > I_2 > \ldots > I_8$. Both the confidentiality level and integrity level are 3-bit binary numbers, as shown in Table 2.

**Table 2.** Secret level.

| Confidentiality Level | Integrity Level | Binary | Secret Level |
|:---:|:---:|:---:|:---:|
| $C_1$ | $I_1$ | 111 | Level-1 |
| $C_2$ | $I_2$ | 110 | Level-2 |
| $C_3$ | $I_3$ | 101 | Level-3 |
| $C_4$ | $I_4$ | 100 | Level-4 |
| $C_5$ | $I_5$ | 011 | Level-5 |
| $C_6$ | $I_6$ | 010 | Level-6 |
| $C_7$ | $I_7$ | 001 | Level-7 |
| $C_8$ | $I_8$ | 000 | Level-8 |

PVMH has 16 security categories or access permissions. The security category set is defined as $K = \{K_1, K_2, \ldots, K_{16}\}$. The security category is a 16-bit binary number, where each bit is a specific access permission. When the $i^{\text{th}}$ bit from the left is set to 1, the subject gains access to the object in $K_i$ mode. When it is set to 0, the subject loses this access.

*4.3. PVMH Model Implementation*

We designed the PVMH architecture according to the security control module of the cloud computing platform, as shown in Figure 1.

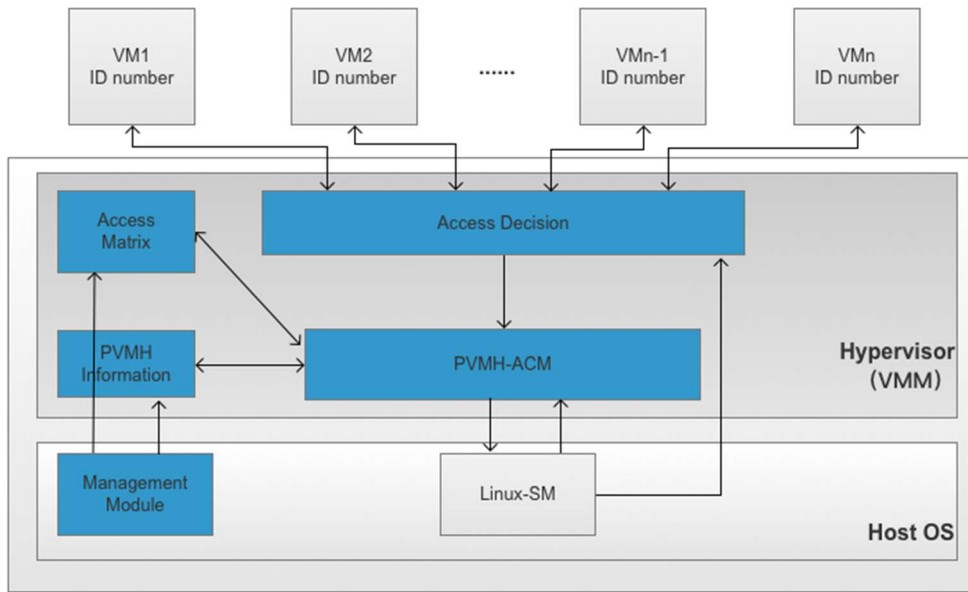

**Figure 1.** The PVMH architecture.

The PVMH framework prototype system is divided into two parts, with the core part in the hypervisor and the other part in the Host OS. The Host OS has an Access Management module. The hypervisor includes an Access Matrix module, an access control module (PVMH-ACM), a PVMH Information module, and an Access Decision module.

1.  Access Management Module: The Access Management module is located in the Host OS, which is the entrance for the system administrator managing the entire PVMH module. When creating a virtual machine, the administrator can set the confidentiality level and integrity level according to specific requirements, and manage both the access matrix and information structure, according to the actual situation.

2.  Access Decision Module: The main task of the Access Decision module is to check the access request sent by the virtual machine, to determine whether the access attribute of the PVMH

module is satisfied, and filter requests that are illegal or have malicious data. Only legitimate requests are sent to PVMH-ACM module.

3.  Access Matrix Module: The Access Matrix module is located in the virtual machine manager and stores all the required access matrices.

4.  PVMH-ACM Module: The PVMH-ACM module is a concrete implementation of the security hook function interface provided by the Linux Security module (LSM). In addition, this module implements the specific functions of the PVMH model.

5.  PVMH Information Module: Each virtual machine has its own information structure (PVMH Information), which is responsible for recording information about running virtual machines. The information includes the ID number of the virtual machine, the confidentiality level, the integrity level and the security category, and the pointer to the entry of the access matrix list entry and the current access set b(s), which was gradually established in the process from virtual machine starting-up to running.

6.  Linux Security Module (LSM): LSM is the basis for implementing the PVMH-ACM module. When the underlying operating system starts, the LSM starts to function. When the corresponding security hook function is called, the user-implemented security module is called immediately.

When the virtual machine issues an access request, the PVMH-ACM module determines whether to allow access to the resource according to the corresponding access control rules. The access control flowchart is shown in Figure 2:

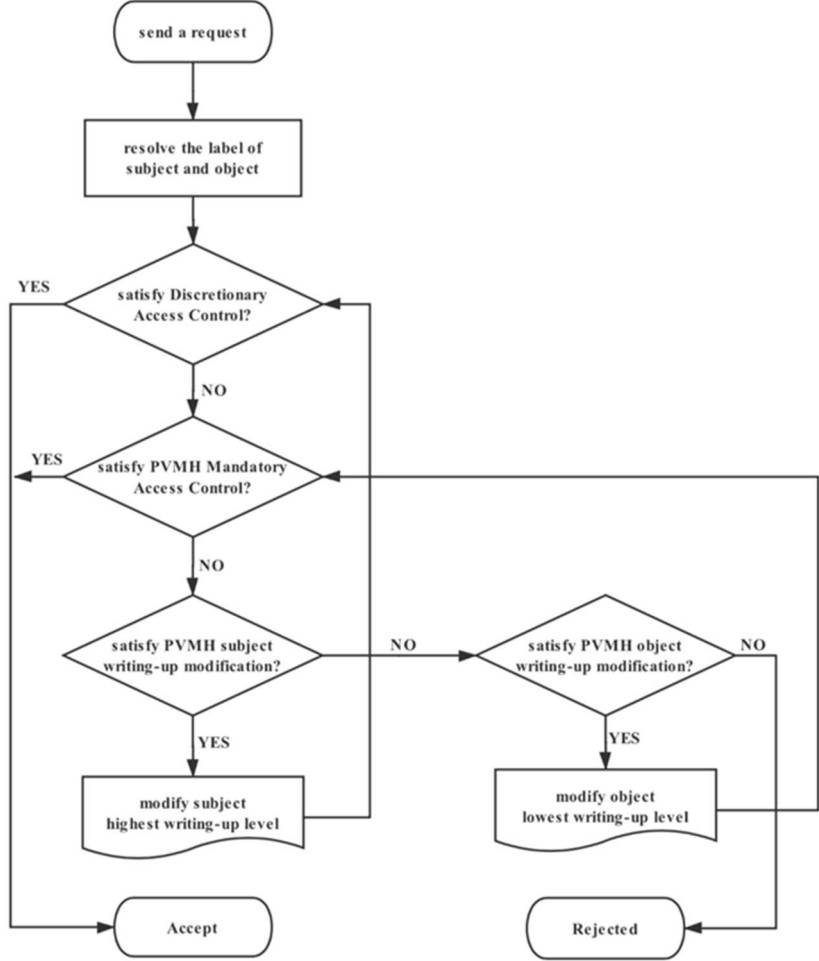

**Figure 2.** Access flowchart. high-res figure

According to PVMH architecture, the specific implementation process is as follow:

**Step 1**: The subject virtual machine sends a request to access the object virtual machine in a certain mode. The Access Decision module intercepts these requests, checks whether they conform to the access attributes of PVMH module, filters the illegal or malicious data requests directly, returns "Error", and sends the requests that conform to the access attributes to the PVMH-ACM module in Hypervisor.

**Step 2**: The PVMH-ACM module queries in the Access Matrix and PVMH Information by resolving the incoming message, and makes decisions according to the state transition rules.

Specifically, it takes it as an example that the subject virtual machine (SID) accesses the object virtual machine (OID) in read-only (R) mode.

(1) The PVMH-ACM module queries the current access set $b(S)$ of the subject virtual machine from the PVMH Information module. If it finds the target pair (OID, R), the PVMH-ACM module accepts the request and returns "yes" directly; otherwise, it jumps to (2) to continue.

(2) All access matrix items associated with SID and OID are stored into a linked list. The PVMH-ACM module searches the linked list from the head node. If it finds the target tuple (SID, OID, 1XXX) (X is 0 or 1), it jumps to (3) for a further decision; otherwise, it rejects the request and returns "no" directly.

(3) The PVMH-ACM module finds the information of the object virtual machine in the PVMH Information module, and compares the confidentiality level of the subject virtual machine with that of the object virtual machine. If rule $PVMH - R_1$ is satisfied, the PVMH-ACM returns "yes" as a decision result and adds the value pair (OID, R) into the current access set $b(S)$; otherwise, it rejects the request and returns "no".

The entire access process is stored in the PVMH Information module, and when the same access is repeated in the future, the PVMH-ACM access control module will directly output the result, instead of matching the subject and object security level and other Information.

**Step 3**: The PVMH-ACM module sends the decision result to the LSM module. If the PVMH-ACM module outputs "yes", the LSM module allows the subject virtual machine to access the object virtual machine, and carries out the security access control according to the specific hook function; otherwise, access is rejected.

## 5. Experiments

### 5.1. Basic Environment

The experiment is performed on a Dell PC with the following configuration, as shown in Table 3.

**Table 3.** Hardware parameters.

| Hardware | Parameter |
| --- | --- |
| CPU | Intel Core i7-6700, 3.4 GHz |
| Memory | 8GB |
| Hard disk | 1TB |

In this experiment, Xen is used as the private cloud platform driven by the virtualization environment, and the "virt-manager" tool is used to manage virtual machines. For convenience, the host operating system is configured with the graphical interface, and since the virtual machine requires only a basic environment, the graphical interface is removed from the guest operating system. The details are shown in Table 4.

**Table 4.** Operating system environment.

| Machine | System Version | Kernel Version | Graphical Interface |
|---------|----------------|----------------|---------------------|
| Host machine | Ubuntu 16.04.5 | 4.15.0-42-generic | Yes |
| Virtual machine | CentOS-6.10 | 4.4.163-1.el6.elrepo.i686 | No |
| Xen Hypervisor | N/A | xen-hypervisor-4.6-amd64 | N/A |

*5.2. The Initialization of PVMH Module*

The PVMH module needs to be initialized before it can run. After initialization, the legal access attributes are stored in the Access Decision module, and the initial access matrix is stored in the Access Matrix module. The system administrator can modify the access matrix according to the access request at any time.

The PVMH-ACM module registers its initialization function through the interface provided by the Linux security module LSM, which is called during the initialization of LSM. The initialization function loads the access matrix into the memory address space of Hypervisor. Based on memory-efficient consideration, such as searching, adding, or deleting elements, an ordered bidirectional linked list is used. In addition, the PVMH-ACM initialization function provides the LSM with information about the security hook functions. The PVMH-ACM module runs at one of these two modes: Mandatory access control or discretionary access control, which depends on the access information returned from the Access Decision module. The PVMH Information module records the information of each virtual machine. PVMH Information is a bidirectional linked list in Hypervisor, and each node holds the relevant information of a specific running virtual machine, which includes the security ID number, confidentiality level, integrity level, security category, and the pointers to the access matrix linked list entry and the current access set b(s). After initialization, only an empty table exists in the PVMH Information module. When a certain virtual machine starts, the corresponding security ID number, confidentiality level, integrity level, and security category are assigned. These four items are read by the PVMH Information module and saved into the bidirectional linked list, which forms the first four elements of this virtual machine.

*5.3. Experiments and Results*

There are many cases of virtual machine hopping attacks. Two of the possible scenarios of virtual machine hopping attacks have been selected here to verify the role of the PVMH module.

**Attack scenario 1:** Virtual machine hopping attacks between virtual machines due to shared memory communication. Typical communication through shared memory is as follows: VM1 creates shared memory and transfers its grant reference to both virtual machines VM2 and VM3; VM2 and VM3 respectively map the authorized memory pages to their respective address spaces; By address mapping, VM2 and VM3 can read or write the shared page as it is exactly in their own memory address. When VM2 and VM3 have finished accessing this shared memory, they revoke the memory page address. At last, VM1 revokes the authorization and reclaims the grant reference. In the experiment, shared memory communication is implemented by dynamic kernel.

Expected result: After the PVMH module is started, if VM2 and VM3 do not satisfy the access control rules, the shared memory cannot be used, thus the dynamic kernel fails and cannot be inserted in VM2 and VM3.

Create virtual machines with the virt-manager tool, as shown in Figure 3.

```
root@work:/exp# virt-install --name VM1 --ram 1024 --vcpus 1 --file images/centos1.img
--file-size 20 --nographics --paravirt --location http://mirrors.163.com/centos/6/os/i3
86/ --extra-args="text console=com1 utf8 console=hvc0"
```

**Figure 3.** Create VM1.

After creation, list all running virtual machines, as shown in Figure 4.

```
[root@work:/exp# virsh list
 Id    Name                        State
--------------------------------------------------
 0     Domain-0                    running
 1     VM1                         running
 2     VM2                         running
 3     VM3                         running
```

**Figure 4.** Command listing all running virtual machines.

The security level and category of virtual machines are given in Table 5.

**Table 5.** Security settings of VM1, VM2, and VM3.

| Subject/Object | SID = OID = ID | Confidentiality Level | Integrity Level | Security Category |
|---|---|---|---|---|
| VM1 | 0000 0000 0000 1 | $C_1(111)$ | $I_2(110)$ | 0011 0100 1000 1010 |
| VM2 | 0000 0000 0001 0 | $C_3(101)$ | $I_5(011)$ | 0000 1011 1001 0110 |
| VM3 | 0000 0000 0001 1 | $C_5(100)$ | $I_5(011)$ | 0010 1011 1010 0011 |

Obviously, VM1 > VM2 > VM3 when comparing confidentiality and VM1 > VM2 = VM3 when integrity is concerned. Without PVMH, VM1, VM2, and VM3 can access each other. After PVMH is configured, however, VM1 can access VM2 and VM3 while VM2 and VM3 cannot access VM1.

The access matrix is given in Table 6.

**Table 6.** Access matrix of VM1, VM2, and VM3.

| Subject/Object | SID=OID | Access Attribute | | | Flag |
|---|---|---|---|---|---|
| | | R | A | W | |
| VM1 | 0000 0000 0000 1 | 1 | 1 | 1 | 1 |
| VM2 | 0000 0000 0001 0 | 0 | 0 | 0 | 1 |
| VM3 | 0000 0000 0001 1 | 0 | 0 | 0 | 0 |

The shared memory is created in VM1 and the key log is printed, as shown in Figure 5.

```
[[root@localhost shared_mem]# insmod dy_dom1.ko domid2=2 domid3=3
[[root@localhost shared_mem]#
[[root@localhost shared_mem]# dmesg | tail -4
[12827.204336] DY: Get free page from kernel, virt: 0xdb566000
[12827.204345] DY: message to share is "Hello, by DY in DOM#1"
[12827.204351] DY: Grant_Ref is 797, input this as dom2.ko param
[12827.204356] DY: Grant_Ref is 798, input this as dom3.ko param
```

**Figure 5.** Create shared memory in VM1.

The function of the kernel in VM1: First take a physical page of 4K size; then write "Hello, by DY in DOM#1" into this page; the starting memory address is 0xdb566000 in the address space of VM1; then authorize according to the ID number of VM2 and VM3 and return the corresponding grant reference identifier, which is 797 for VM2 and 798 for VM3.

According to the ID number of VM1 and the grant reference identifier mentioned above, the shared memory is referenced in the VM2 through dynamic kernel. Without PVMH, VM2 successfully reads the shared information written by VM1, as shown in Figure 6.

```
[root@localhost shared_mem]# insmod dy_dom2.ko gref=797 domid=1
[root@localhost shared_mem]#
[root@localhost shared_mem]# dmesg | tail -3
[11680.099662] DY: init_module with gref = 797, domid = 1
[11680.100828] DY: shared_page = e19c0000, handle = 5, status = 0
[11680.100837] DY: info from dom1: Hello, by DY in DOM#1
```

**Figure 6.** Refer shared memory in VM2, without PVMH module.

The function of the kernel in VM2: First, the 4K size is divided from the address space for mapping the shared page. Then VM2 maps the shared page to its own address space according to the ID number of VM1 and the grant reference identifier; in VM2, the starting address of this mapped page is 0xe19c0000; after that, the information "Hello, by DY in DOM#1" written by VM1 can be read and the access is completed.

In order to verify the role of the PVMH module, the dynamic kernel needs to be removed from VM2 at first, then the PVMH module enabled and the dynamic kernel reinserted in VM2, as shown in Figures 7 and 8 respectively.

**Figure 7.** Enable the PVMH module.

**Figure 8.** Refer shared memory in VM2, with PVMH enabled.

After starting the PVMH module, VM2 loses its permission to access the shared memory, which results in the failure of the dynamic kernel insertion. Similar results can be observed in VM3 with and without the PVMH module. This is consistent with PVMH rules, because VM2 has lower confidentiality and lower integrity than VM1.

**Attack scenario 2:** The attacker uses the VM4 to mount and modify the /boot partition of VM5 through Hypervisor. As a result, VM5 cannot be started, causing the virtual machine hopping attack.

Expected result: After the PVMH module is started, if the access control rule is not satisfied, VM4 cannot access the/boot partition of VM5, even using the privileged virtual machine Dom0.

Set the security level and category of VM4 and VM5, as shown in Table 7.

**Table 7.** Security settings of VM4 and VM5.

| Subject/Object | SID = OID = ID | Confidentiality Level | Integrity Level | Security Category |
|---|---|---|---|---|
| VM4 | 0000 0000 0010 0 | $C_4$(100) | $I_6$(010) | 0011 0000 1010 1000 |
| VM5 | 0000 0000 0010 1 | $C_3$(101) | $I_5$(011) | 0000 0011 1001 1000 |

Obviously, VM4 < VM5 when comparing both confidentiality and integrity. Without PVMH, VM4 can mount the/boot partition of VM5 through Dom0. After PVMH is started, however, VM4 cannot access VM5 anymore.

The access matrix is given in Table 8.

**Table 8.** Access Matrix of VM4 and VM5.

| Subject/Object | SID = OID | Access Attribute | | | Flag |
|---|---|---|---|---|---|
| | | R | A | W | |
| VM4 | 0000 0000 0010 0 | 1 | 0 | 1 | 1 |
| VM5 | 0000 0000 0010 1 | 1 | 0 | 0 | 1 |

VM4 uses the privileged virtual machine Dom0 to view the partition of VM5, as shown in Figure 9.

```
root@work:/exp# fdisk -l images/centos5.img
Disk images/centos5.img: 20 GiB, 21474836480 bytes, 41943040 sectors
Units: sectors of 1 * 512 = 512 bytes
Sector size (logical/physical): 512 bytes / 512 bytes
I/O size (minimum/optimal): 512 bytes / 512 bytes
Disklabel type: dos
Disk identifier: 0x000b02b1

Device              Boot    Start       End  Sectors  Size Id Type
images/centos5.img1 *        2048   1026047  1024000  500M 83 Linux
images/centos5.img2       1026048  41943039 40916992 19.5G 8e Linux LVM
```

**Figure 9.** Partition of VM5.

The disk of VM5 is divided into two partitions: /and/boot. Before the PVMH module is enabled, VM4 mounts and accesses the/boot partition of VM5 through Dom0. The "Start" value of the partition/boot, 2048, is used to calculate the offset in command mount, as shown in Figure 10:

```
root@work:/exp# mount -o loop,offset=$((2048*512)) images/centos5.img /mnt/
root@work:/exp#
root@work:/exp# ll /mnt/
total 33826
dr-xr-xr-x.  5 root root     1024 Dec 14 16:49 ./
drwxr-xr-x 25 root root     4096 Dec  7 14:51 ../
-rw-r--r--.  1 root root   170974 Dec 14 00:16 config-4.4.167-1.el6.elrepo.i686
drwxr-xr-x.  3 root root     1024 Nov 21 10:50 efi/
drwxr-xr-x.  2 root root     1024 Dec 14 16:49 grub/
-rw-------.  1 root root 25086469 Dec 14 16:39 initramfs-4.4.167-1.el6.elrepo.i686.img
drwx------.  2 root root    12288 Nov 21 10:46 lost+found/
-rw-r--r--.  1 root root   314241 Dec 14 00:17 symvers-4.4.167-1.el6.elrepo.i686.gz
-rw-------.  1 root root  2925811 Dec 14 00:16 System.map-4.4.167-1.el6.elrepo.i686
-rwxr-xr-x.  1 root root  6109216 Dec 14 00:16 vmlinuz-4.4.167-1.el6.elrepo.i686*
```

**Figure 10.** Mount and view the/boot partition of VM5, without PVMH.

Since VM4 can modify the/boot partition of VM5, it can attack VM5 easily. Unmount the partition, enable the PVMH module and remount the/boot partition of VM5. The results are as given in Figure 11.

```
root@work:/exp# mount -o loop,offset=$((2048*512)) images/centos5.img /mnt/
mount: mount(2) failed: No such file or directory
root@work:/exp#
root@work:/exp# ll /mnt/
total 8
drwxr-xr-x  2 root root 4096 Jul 31 08:30 ./
drwxr-xr-x 25 root root 4096 Dec  7 14:51 ../
```

**Figure 11.** Mount and view the/boot partition of VM5, with PVMH enabled.

After the PVMH module is enabled, VM4 cannot mount the/boot partition of VM5. It should be noted that the mount operation corresponds to a series of Linux system calls. In our implementation, after the PVMH module interception, it affects the input of subsequent system calls, so the error message is "No such file or directory".

Through these two specific scenario experiments, it can be seen that the PVMH module has played a role in preventing virtual machine hopping attacks. In addition to the above experimental results, performance cost testing of PVMH is required to understand the impact of integrating PVMH into the original virtualization platform.

Performance overhead experiment: Taking attack scenario 1 as an example, the most critical and time-consuming step in shared memory communication is to generate a system interruption through the HYPERVISOR_grant_table_op function, and then call a set of hypercalls. Before and after the

PVMH module is enabled, the actual time of the function call is tested separately to measure the performance impact of the PVMH module on the original virtualization platform.

As shown in Figure 12, without PVMH module, it takes 853 microseconds to call a set of hypercalls via HYPERVISOR_grant_table_op. After loading the PVMH module, the average time is 932 microseconds, which results in an additional time loss of 9%.

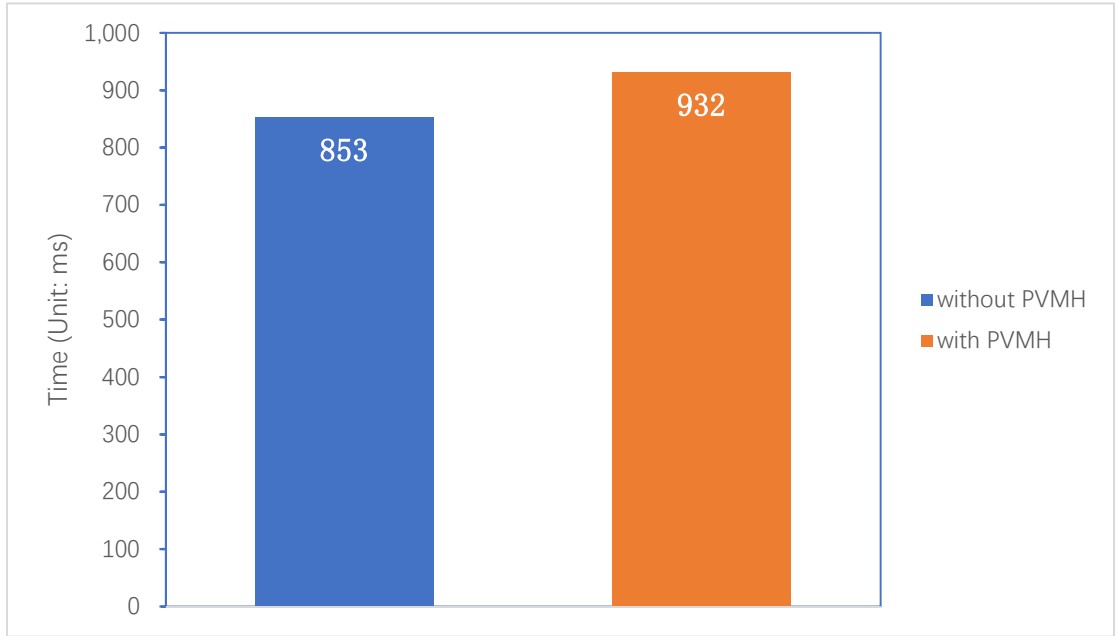

**Figure 12.** Performance with and without the PVMH module.

It can be seen from the above analysis that the PVMH module can effectively prevent the virtual machine hopping problem in the cloud computing environment without significantly reducing the system performance.

## 6. Conclusions

By analyzing the above experimental results, it is clear that PVMH can effectively prevent Virtual Machine hopping attacks and ensure the security between different virtual machines on the same host. Since the PVMH module needs to call the system function when it is running, it consumes a certain amount of system performance, but for the overall effect, the additional loss is within an acceptable range.

In future research, the safety rules of PVMH could be further streamlined, making the PVMH model more suitable for preventing Virtual Machine hopping attacks. In addition, we need to strengthen the connection between the PVMH module and the LSM module, which could reduce the workload and achieve better preventive effects.

**Author Contributions:** Ying Dong and Zhou Lei conceived and designed the PVMH model; Ying Dong performed the experiments and wrote the paper; Zhou Lei revised the paper.

**Funding:** This research received no external funding.

**Conflicts of Interest:** The authors declare no conflict of interest.

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
