# Peer review of "An Access Control Model for Preventing Virtual Machine Hopping Attack"

_futureinternet, doi:10.3390/fi11030082_

Reviewer 1 Report

The authors present design and development of the access control model called PVMH to prevent VM Hopping attacks based on the BLP14 model and the Biba model. Generally, the problem addressed is significant and interesting. The experiments show that PVMH module can effectively prevent the virtual machine hopping problem in the given scenarios without significantly reducing the system performance.  However, the authors need to satisfy by giving more information for the following issues to make the paper self-contained.

1. The attacks on VM discussed in the introduction is too generic, more specific details about the VM hopping attack in cloud platform and its countermeasures are required. The motivation for efficient access control is thus sound and convincing.

2. The discussion about the security characteristic of PVMH model using a more strict integrity restriction based on the security characteristic of BLP model is unclear. The authors need to give more information on such security property.

3. How can we ensure that the request sent to access decision module is illegal? 

4. Include and cite more recent papers into the reference list. 

Author Response

Our response is uploaded as PDF file.

Reviewer 2 Report

This paper proposed PVMH, an access control model to prevent VM hopping attack. The model is designed to be based on BLP and Biba models. The PVMH is implemented on Xen and evaluated with two attack scenarios and also show reasonable performance overhead. 

Things need to improve:

If possible, maybe the authors can share more real-world examples of VM hopping attack to better motivate the work. VM hopping attacks can be caused by various problems such as vulnerability, side-channel, mis-configurations. Different cases may require different defenses. It is unclear to what extent of defense PVMH can achieve, and how PVMH is different from existing work. A more detailed related work comparing previous work, real-world attacks and PVMH can be helpful.

In Section 4, the paper introduces the design of PVMH. However, it is unclear what the intuition and principle behind the design of PVMH. In particular, Section 4.1.2 and 4.1.3 need more detailed explanation on how these rules work. Are these rules targeting the same or different attack scenarios? If so, what are those specific attack scenarios? The author can give some examples to illustrate the design of these rules. In addition, the notations of the rules are not quite clear. 

The paper should also clearly state what the Trusted Computing Base (TCB) this paper assumes. To design an access control system, it needs a trusted way to enforce it. If the host system is trusted, then it can be used to deploy and enforce the access control policy. However, if the host system also has vulnerabilities and can be compromised, then the access control system itself can be attacked and become invalid. 

In Section 5, the two attack scenarios can be explained in more details. It seems that the paper assumes that there are some vulnerabilities existing either in VM or Hypervisor. These assumptions, TCB and experiment settings need to be listed clearly.

Author Response

Our response is uploaded as PDF file.

Round  2

Reviewer 1 Report

The authors have addressed my comments thoroughly. The paper is now recommended for the publication.